# Pros and Cons of Alternatives to Piglet Castration: Welfare, Boar Taint, and Other Meat Quality Traits

**DOI:** 10.3390/ani9110884

**Published:** 2019-10-30

**Authors:** Michel Bonneau, Ulrike Weiler

**Affiliations:** 1IFIP, The French Pork and Pig Institute, La Motte au Vicomte, B.P. 35104, 35 651 Le Rheu CEDEX, France; 2Department of Behavioral Physiology of Livestock 460f, University of Hohenheim, 70599 Stuttgart, Germany; ulrike.weiler@uni-hohenheim.de

**Keywords:** pig, boar taint, meat quality, welfare, castration

## Abstract

**Simple Summary:**

Each alternative to traditional surgical castration has its pros and cons. Depending on the societal context, the production system, and the target market(s), pork supply chains may choose the alternative(s) that best fit(s) their situation. Conflicting aims occur between animal welfare issues and the efficiency of production, whereas product quality and welfare issues are mostly synergic.

**Abstract:**

This paper reviews the pros and cons of various alternatives to the surgical castration of male piglets without pain relief. Castration is mostly motivated by the presence of boar taint in the meat from some entire male pigs. It results in pain during surgery and markedly increases feed costs and the fat content of the carcass. Raising entire male pigs avoids pain at castration, but animals can suffer from increased stress during the finishing period because of aggressive and mounting behavior. Feed efficiency and carcass quality are much better than in surgical castrates. The quality of meat from entire male pigs is lower because of boar taint, a reduced intramuscular fat content, and increased unsaturation of the fat. Immunocastration prevents boar taint, pain associated with surgery, and stress related to aggressive and mounting behavior. Feed efficiency and carcass quality are intermediate between surgical castrates and entire males. Meat quality is similar to surgical castrates. Anesthesia alone prevents pain during surgery, but not after, while analgesia alone mitigates pain after surgery, but not during it. With the currently available methods, the cost of combined anesthesia and analgesia is too high for conventional production systems in most countries.

## 1. Introduction

The surgical castration of male piglets has been a traditional practice for ages and is still common in most countries. This procedure is motivated by the presence of boar taint in the meat from some entire male pigs. Even if some countries in Western Europe have promoted the use of anesthesia or analgesia, the procedure is still often practiced without any pain relief and is therefore facing increasing criticism because of the pain inflicted to the animal as a consequence of the surgery [1,2]. To account for that, in 2010, a number of European stakeholders committed themselves to stopping surgical castration by 2018, provided that satisfactory solutions are found to the various challenges associated with the production of entire (uncastrated) male pigs. Alternatives to surgical castration without pain relief have been developed and are implemented in some countries. However, 75% of male pigs are still surgically castrated in the EU [3,4]. Indeed, none of the available alternatives are fully satisfactory. Moreover, there are still some countries, especially in Eastern Europe, where most stakeholders consider that the surgical castration of male pigs without pain relief is not an issue. Depending on the constraints of the local context, the advantages and drawbacks of each alternative must be carefully considered. The COST action Innovative Approaches for Pork Production with Entire Males (IPEMA), which has been running since 2017, aims to raise awareness of the issue and bring scientists and stakeholders together “to find general, region-specific or chain-specific solutions to facilitate the development of alternatives to surgical castration of piglets” [5].

## 2. Why Are Piglets Castrated?

The main reason for castrating male pigs is the occurrence of boar taint, an offensive odor and flavor perceived when cooking and eating the meat from some entire male pigs. Two main compounds have been demonstrated to be associated with boar taint: androstenone and skatole [6]. Because these compounds are lipophilic, they accumulate in the adipose tissue of growing animals in relation to pubertal development. In carcasses where the concentration of compounds is higher than the individual sensitivity threshold, sensitive consumers can perceive the cooking odor or flavor of meat as unpleasant [7].

Androstenone (5α-androst-16-ene-3one) is a testicular steroid with a urine-like smell [8]. Its production in the Leydig cells is regulated by the hypothalamic-pituitary-gonadal axis, in the same way as the synthesis of the gonadal hormones androgens and estrogens [9]. After being released in the blood, androstenone can be catabolized by the liver, stored reversibly in the adipose tissue, or taken up by the salivary glands, where it is reduced to α-androstenol and β-androstenol [10] that are excreted in saliva, where they act as pheromones to induce puberty in gilts or elicit mating behavior in the sow. Androstenone levels in the fat of entire male pigs range from 0.1 to 0.2 µg/g to 5 to 10 µg/g, according to a lognormal distribution [11]. The human sensitivity to androstenone is highly variable. About one third of consumers are anosmic to androstenone (cannot smell it), whilst another third are highly sensitive and reject pork with already low androstenone concentrations [12,13]. The remaining third of consumers also perceive the odor, but consider it as pleasant [7,14].

Skatole (3-methyl-indole) is a metabolite of the amino acid tryptophan, with a fecal odor [15]. It is synthesized in the colon by microbial degradation of the indigestible but fermentable portion of the feed and intestinal cell debris. Skatole is absorbed from the large intestine and circulates in the blood, where it can be catabolized by the liver or stored reversibly in the adipose tissue. The main reason why entire male pigs have higher skatole levels in adipose tissue than barrows or gilts is that the hepatic degradation of skatole is reduced, due to inhibition of the activity of catabolic enzymes by androstenone, testosterone, or 17β-estradiol [16,17,18,19]. Skatole levels in the fat of entire male pigs range from 0.01 to 0.02 µg/g to 0.5 to 1.0 µg/g, according to a lognormal distribution [11]. The high variability in the human perception of androstenone odor does not exist for skatole: most consumers dislike the odor and flavor of meat exhibiting high levels of skatole [7,20].

## 3. Consequences of Surgical Castration

### 3.1. Unfavorable and Favorable Consequences of Surgical Castration for Animal Welfare

Surgical castration results in pain for the male piglets, both during and after the surgery [1,2]. This is demonstrated by high-frequency vocalization (screams); behavioral resistance; and increases in the heart rate, adrenaline, noradrenaline, and cortisol levels and expression of the c-fos protein in neurons of the spinal cord. The animals also show more pain-related behavior. There is some evidence of health impairment in castrated compared to entire male pigs, leading to higher mortality in surgically castrated piglets than in intact males [21].

The intense but short duration of the surgery-related pain in piglets is a clear negative aspect of surgical castration. However, castration also has positive aspects regarding animal welfare. Indeed, it avoids the expression of mounting and aggressive behaviors observed in the more restless entire males, resulting in long-lasting reduced welfare for the dominated animals that are harassed by their dominant pen mates [22]. It also avoids penile injuries that are quite common in entire males [23,24].

### 3.2. Consequences of Surgical Castration for Feed Efficiency, Carcass Content, and Meat Quality

The surgical castration of male piglets induces an increase in daily feed consumption with no compensation in growth rate. This results in a sharp reduction in feed efficiency: a total of 10% to 15% more feed is required to produce the same amount of meat compared to boars and nitrogen excretion is about 15% higher than in entire pigs [25]. This results in a sharp increase in feeding costs and environmental impacts.

The effect of surgical castration on a number of carcass and meat quality traits was reviewed in 2009 [25] and quantified in 2012 in a meta-analysis of 28 published studies [26]. Compared with entire males, castrates exhibit a higher killing out percentage, but a markedly augmented fat content in the carcass, which results in a lower selling price for the carcass. The intramuscular fat content is higher in castrates than in entire males, which is favorable for eating quality. Some experiments have reported higher ultimate pH values and a lower frequency of Dark Firm Dry (DFD) meat in castrates than in entire males, in connection with their lower activity, which is also favorable for meat quality. The fatty tissue of castrates contains less water, and more saturated and less polyunsaturated fatty acids, which makes castrates’ fat firmer and less prone to becoming rancid during the storage and maturation of dry cured products. More saturated fat is, however, less healthy [26,27].

In summary, early surgical castration avoids boar taint accumulation. It also has a number of other advantages. It prevents undesirable male aggressive and sexual behavior during the fattening period, so that barrows are quieter and easier to manage than entire males. Meat and fat quality are also better in castrates than in entire males. The disadvantages of surgical castration are the labor cost to perform castration, reduced welfare related to pain during castration, the higher feeding cost and impact on the environment, and the reduced value of the carcass because of the elevated fat content.

## 4. What Are the Alternatives?

The technically available alternatives to surgical castration without pain relief have been reviewed in a collective report prepared for the European Food Safety Authority [28]. They include (i) sperm sexing to produce only females, (ii) the injection of chemical compounds into the testes to destroy the tissue, (iii) the administration of exogenous hormones to inhibit the hypothalamic-pituitary-gonadal axis, (iv) surgical castration with pain relief, (v) immunocastration, and (vi) entire male production.

For various reasons, the first three alternatives cannot be realistically considered [1,2]. Contrary to the situation in bovine, sperm sexing in pigs is extremely tedious, inefficient, and expensive in porcine; the injection of chemicals into the testis results in swelling and pain for the animal; and the administration of exogenous hormones is not legally permitted in the EU. This paper will examine, in detail, the advantages and disadvantages of the three remaining alternatives: the production of entire males, immunocastration, and surgical castration with pain relief.

## 5. Entire Male Pigs

### 5.1. Pros and Cons of Raising Entire Male Pigs

Extensive information on the use of entire male pigs is available in recent reviews [29,30]. The advantages and disadvantages of surgical castration, as described in Section 3 above, are reversed to disadvantages and advantages, respectively. The pros of raising entire males include the avoidance of a cumbersome job and pain related to surgical castration; the reduction of feed costs and impact on the environment; the increase in muscle content; and the increase in unsaturated fat, which is healthier. The cons of raising entire males include the difficulties experienced by some farmers in managing the more restless entire males; impaired animal welfare for the animals harassed by their dominant pen mates exhibiting mounting and aggressive behavior; penile injuries; the lower meat quality in relation to the reduced intramuscular fat content and more frequent occurrence of DFD meat; increased fat unsaturation, which is detrimental for processing dry-cured products [31]; and, last but not least, boar taint, which is a serious risk for consumer satisfaction [32]. The detection of boar taint on the slaughter line and inclusion of the tainted meat in lower-value processed products (see Section 5.2.2 and Section 5.2.3 below) represent additional costs for the chain. Practice-based ways of dealing with management problems specific to entire males are provided by the European Commission (2019) [30].

### 5.2. Management of the Boar Taint Problem

Management of the boar taint issue requires an integrated approach all along the supply chain. Genetic, nutritional, and management factors can be used to reduce the occurrence of animals exhibiting boar taint; sorting methods can be used to detect tainted carcasses on the slaughter-line; and various processing methods can be implemented to inhibit the perception of boar taint in processed products. This section will only present a quick summary of the issue. For more detailed information, the reader can refer to reviews [25,29] and a recently published document on best practices [30].

#### 5.2.1. Decreasing the Occurrence of Animals Exhibiting Boar Taint

Because the occurrence of boar taint is related to sexual development, it tends to increase with the age and weight of animals at slaughter, but the relation is loose and complex because many more factors are involved in the control of boar taint. Androstenone-related boar taint is mostly controlled by genetic factors and skatole-related taint is also under some genetic influence. Genetic selection for low boar taint levels has already been included in some breeding programs and sire lines of “low boar-taint” boars are already in use in practice. However, it is in the dam lines that the selection is most needed (because they exhibit higher androstenone levels than sire lines) and, unfortunately, also the most difficult, because of interdependence with the regulation of fertility traits. New techniques and strategies are promising, but do not provide a rapid solution [33,34,35].

Skatole can be efficiently controlled by feeding measures, such as the addition of non-digestible/fermentable feedstuffs to the diet for a few days/weeks before slaughter [36]. Low-protein diets are less efficient for reducing skatole, but are also considered because they are much cheaper. Management methods resulting in cleaner and less stressed animals are useful for reducing skatole. However, these measures are not efficient for controlling androstenone levels [37] and thus do not guarantee boar taint-free populations. Finally, transport and lairage conditions before slaughter may affect the production, storage, and catabolism of boar taint compounds, so care must be taken to avoid spoiling the above-mentioned management steps by inappropriate handling and transport [38].

#### 5.2.2. Detection of Boar Taint on the Slaughter Line

Boar taint detection on the slaughter line allows carcasses with boar taint to be sorted out in order to take them out of the fresh pork market and use them for processing into “boar taint-resilient” products (see Section 5.2.3 below). Sorting methods based on olfactory detection by human experts (“human nose” methods) are routinely used in slaughter houses, particularly in those countries where entire male pig production has only recently developed (see Section 8 below). The human nose methods are cheap, but their efficiency for protecting consumers from dissatisfaction is not satisfactorily documented in scientific publications [39]. Much work has been done during the last 20 years to develop instrumental methods, which have the advantage of being objective, without much success [40]. However, a few methods have recently been announced, which seem to be close to reaching the market, and could be implemented at a reasonable price in the range of 1–2 euros per controlled carcass [41,42,43]. Because the meat included in taint-resilient products is less valorized than fresh meat, the percentage of sorted-out carcasses must be low (in the range of 4%–5%) for the whole process to be economically sustainable.

#### 5.2.3. Processing Tainted Meat to Reduce Boar Taint Perception

Reviewed information on the effect of processing on boar taint perception can be found in a 2009 review [25] and in a recent paper on best practices [30]. Briefly, boar taint perception is affected by consumption/preparation temperature (products consumed cold are more taint-resilient than those consumed warm and those cooked at home are more at risk), the amount of tainted fat (tainted meat can be diluted with untainted meat in minced products), and the presence of masking ingredients (smoke is the most documented masking ingredient, but there are other efficient ones). There is therefore a gradient of products from the most taint-resilient ones (minced products consumed cold that include masking ingredients) to the least taint-resilient ones (high-fat products with no masking agent, cooked at home, and consumed warm).

## 6. Immunocastration

### 6.1. The Effects of Immunocastration

A vaccination against GnRH was proven to be effective in preventing boar taint as early as 1986 [44], opening the way to the development of a commercial vaccine [45]. Reviews [46,47] and meta-analyses [26,27,48] of the available information on immunocastration are available. The principle is that a vaccine based on a GnRH construct is administered to the animal to elicit the production of anti-GnRH antibodies. GnRH, being neutralized by the antibodies, does not stimulate LH and FSH production, with the result that testicular development and steroid production in the Leydig cells are stopped. Two shots of anti-GnRH vaccination are required to effectively stop sexual development and decrease boar taint. The first one, administered at around 8–12 weeks, is only a primer. From a few days after the second vaccination, usually performed 4–6 weeks before slaughter, the animals behave like castrates, with a sharp decrease in aggressive and mounting behavior and a marked increase in feed intake to levels higher than those observed in surgical castrates at the same stage of growth [49,50,51,52,53]. Provided that both vaccinations are properly administered, the percentage of non-responders is very low and the occurrence of animals with levels of androstenone or skatole resulting in boar taint is also very low. However, in the conditions of the practice, the likeliness that one of the shots is not properly administered is not negligible, so there are commonly some animals that continue to behave like entire males and exhibit boar taint at slaughter.

Muscle development benefits from the presence of the anabolic hormones androgens and estrogens until the second vaccination. After that, fat development is greatly stimulated. Because of the over feeding taking place after the second vaccination, the overall growth rate during the growing-finishing period is generally higher in immunocastrates than in both entire males and surgical castrates. The resulting feeding costs and carcass quality are intermediate between those observed in entire males and castrates. The longer the delay between the second vaccination and slaughter, the closer the performance is to that of castrates’ [54]. Because the vaccination is somewhat reversible, three shots are required in production systems where the animals are slaughtered at a heavy weight [55,56,57].

### 6.2. Pros and Cons of Immunocastration

Immunocastration is less painful to the animal than surgical castration without pain relief [1], even though some mild adverse reactions can be observed at the immunization site [45,58]. After the second vaccination, the pigs do not display aggressive and mounting behaviors typical of the entire males, which is also beneficial for the welfare of animals. The economic advantages derived from the period before the second shot, when the animals are biologically like entire males, are less important than in entire males but are still there, unless the second vaccination is performed a long time before slaughter. [30,47]. The meat quality (muscle and fat) is close to that of surgical castrates. In particular, the intramuscular fat content is usually closer to surgical castrates than to entire males, which is favorable for meat quality. The cost of the two (or three) shots of vaccine added to the labor costs for performing the vaccinations is a disadvantage. The second vaccination, being performed in quite heavy animals, can be laborious. This is even worse when a third vaccination is required in heavy pigs. Some monitoring has to be done after the second vaccination to detect non-responders on the basis of their behavior and testes size: this represents some supplementary labor. Finally, the risk of self-injection for the workers, although very unlikely, can also be considered as a disadvantage of the procedure [47].

## 7. Surgical Castration with Anesthesia and/or Analgesia

The use of anesthesia and or analgesia to prevent or mitigate pain during surgical castration was reviewed in 2006 [1], 2011 [59], and more recently in 2016, in the final report of the EU-funded project CASTRUM [60]. Useful information can also be found in a recently published document on best practices [30].

### 7.1. Anaesthesia

#### 7.1.1. General Anesthesia

General anesthesia for piglet castration is administered via inhalation (CO_2_/O_2_, Isoflurane) or intramuscularly (Ketamine). For CO_2_/O_2_ anesthesia, piglets inhale a mixture of 70% CO_2_ and 30% O_2_ for at least 30 s and castration is performed within one minute. CO_2_/O_2_ anesthesia is cheap but aversive to the piglets, which experience some pain and discomfort [1,2]. Its efficacy in reducing pain immediately after surgery is also being questioned. Finally, the safety margin between the dose necessary to elicit unconsciousness and the lethal dose is limited. Isoflurane anesthesia needs expensive equipment to give piglets a mixture of isoflurane and air (or O_2_) for at least 90 s. Isoflurane is a potent greenhouse gas and can therefore affect the environment; it can also affect workers, who sometimes report headaches and dizziness when performing the anesthesia. General anesthesia can also be achieved via the intramuscular injection of a mixture of ketamine and azaperone. The use of ketamine is severely restricted because of its hallucinogenic properties. Moreover, because the recovery time from ketamine anesthesia is long, the animal has an increased risk of being crushed by the sow, and also experiences hypothermia. Thorough monitoring is therefore needed to avoid mortality.

#### 7.1.2. Local Anesthesia

Local anesthesia consists of the injection of a local anesthetic into the spermatic cord or the testes. The most commonly used drug is lidocaine. The addition of adrenaline reduces bleeding and extends the duration of anesthesia. Procaine has also been used, but it has a slower onset and shorter duration. Local anesthesia seems to be effective in some studies if carefully performed to avoid pain during the injection and the timing between injection and surgery is correct, but is not effective in all studies for significantly reducing the pain of castration and the positive effect of local anesthesia with lidocaine on piglet welfare during castration is relatively limited [61,62,63]. Association with analgesia is recommended.

### 7.2. Analgesia

The most commonly used drugs for analgesia include Meloxicam and Flunixin (non-steroidal anti-inflammatory drugs) and Metamizole (non-opioid pyrazolone derivative) [60]. Depending on the local regulatory context, analgesia can be administered by the farmers themselves or by veterinarians. Analgesia administered prior to anesthesia improves the efficiency of pain reduction during surgery and its duration after surgery. Analgesia alone does not reduce pain during surgery, but is effective in mitigating pain after surgery. However, the CASTRUM report states that “Potential long acting pain reducing drugs that are effective during and after castration are currently not available” [60].

### 7.3. Pros and Cons of Surgical Castration with Pain Relief

Anesthesia (general or local) is effective for preventing pain during castration, but not for relieving post-operation pain. Conversely, analgesia is effective post-surgery, but not during it. Only combined anesthesia and analgesia is fully effective for avoiding pain, but it is a costly procedure, especially if veterinarians are required. Operational costs per male pig submitted to castration have been calculated as follows: analgesia alone performed by the farmer 0.3 €; analgesia alone performed by a veterinarian 0.7 €; anesthesia with CO_2_/O_2_ performed by the farmer 0.5 €; local anesthesia performed by the farmer <1 €; local anesthesia performed by veterinarians 2 €; anesthesia with isoflurane performed by the farmer 1.3 €; and anesthesia with isoflurane combined with analgesia performed by a veterinarian 4 € [64,65,66].

Surgical castration with pain relief has all the advantages of surgical castration (easier management, lower expression of aggressive and mounting behavior, absence of boar taint, better fat quality), plus improved welfare (no or less pain during surgery). It also has all its disadvantages (increased feeding costs due to lower feed efficiency, higher environmental impact, higher fat content), plus increased costs associated with the application of pain relief.

## 8. The Current Situation in Europe

The current situation in Europe regarding the use of surgical castration and its various alternatives was monitored in 2008 [67] and more recently in 2016 [4], and reviewed in 2015 [29]. Relevant information can also be found in a recent document reviewing the best practices in the field [30].

### 8.1. Entire Male Production

Farmers have been raising only entire males since the sixties in the UK and Ireland to benefit from their much higher efficiency for producing lean meat. At that time, slaughter weights were much lower in these countries than in continental Europe and the occurrence of boar taint was consequently very low. Although slaughter weights in the UK and Ireland have increased substantially since that time, almost 100% of the males are still currently left entire in these countries. Castration has also been abandoned in Spain, Portugal, and Greece in mainstream standard production, for the same reasons. Surgical castration is, however, still performed to some extent in these countries, particularly in the high-quality production systems where the animals are slaughtered at a much higher slaughter weight when the occurrence of boar taint is much more likely. Most high-quality products also require large quantities of unsaturated fat, which are more readily obtained from castrates.

Until the beginning of the 21st century, surgical castration without pain relief was the standard in a vast majority of the other European countries, including all major pork producing countries, with the exception of Spain. During the last 10–15 years, societal and market pressure has induced pork production chains to change their practice in the western part of Europe. The first move towards entire male production was observed in the Netherlands and Belgium, under strong pressure from the market, which was itself under high pressure from animal right NGOs. In the Netherlands, about 70% of the males are now left entire, but the market is reluctant to accept more. Only 15% of the males are left entire in Belgium, where immunocastration is more common (see Section 8.2 below). In France and Germany, 20% of the males are not castrated: a handful of big companies have abandoned castration to a large extent, whereas the remaining companies are still sticking to surgical castration, using analgesia to mitigate pain (see Section 8.3 below).

### 8.2. Immunocastration

Outside Europe, immunocastration has been used on a large scale during the last two decades in Australia and New Zealand and has recently developed strongly in South America, particularly in Brazil. Its development in Europe is still impaired by a strong reluctance from chain actors, based on assumed rejection of the practice by the consumers. The main concern is that the consumption of pork from immunocastrates might affect human fertility. The safety for consumers, however, is well-documented [53,68]. The antigenic GnRH fragment of the vaccine only has a potency of 0.2% on LH-release when compared to injections of the decapeptide. The carrier protein is also used for other vaccines and has not demonstrated toxic or hormonal activity. The construct of GnRH-fragment conjugated to the carrier protein has not exhibited hormonal activity, irrespective of whether it is administered orally or injected. Belgium is, so far, the only European country where immunocastration is practiced to a substantial degree (15% of the males). This is the result of a decision made by a couple of retailers, who favor this option rather than entire males. In other countries, immunocastration can be found on a low scale (<10%) in Sweden, the Czech Republic, Slovakia, Romania, Italy, and Spain, either as an answer to the societal demand to stop castration or to benefit from the economic advantage of immunocastrates compared to surgical castrates. Immunocastration is also under consideration and experimentation in production systems using heavy pigs for high-quality products.

### 8.3. Continuation of Surgical Castration

Pork production in Italy is unique in that it uses very high slaughter weights. Entire males at such weights are all sexually mature and would consequently exhibit a high risk of boar taint and display aggressive and mounting behavior to a large degree. Moreover, entire males provide less fat that is more unsaturated, which is unfit for the production of the high-quality dry cured products that are typical of Italy. For all these reasons, Italian stakeholders are very reluctant to abandon surgical castration.

In most Eastern countries, piglet castration is not yet considered an issue: surgical castration without any pain relief is still the standard and there is so far no real pressure from society or the market to change the practice [69]. Moreover, high-quality products with high fat contents, which require saturated fat, are also common in those countries [60,69].

Since 2002, local anesthesia performed by veterinarians has been mandatory in Norway. In other countries, local anesthesia can be applied by the farmers. Since 2010, general anesthesia with isoflurane combined with analgesia is mandatory for performing piglet castration in Switzerland, where farmers can apply it themselves provided they have received special training. Similar regulations have recently been released for Germany. In other countries, general anesthesia with isoflurane can only be performed by veterinarians. In the Netherlands, farmers who are still practicing surgical castration use anesthesia with CO_2_/O_2_. Many Western countries use analgesia alone on a large scale. Intensive studies are currently being carried out in Germany to find easier and cheaper methods for efficient pain relief during and after castration.

## 9. Conclusions

Each alternative has its pros and cons and there is no worldwide or European-wide best solution. Depending on the constraints and opportunities presented by the societal context, the production system, and the target market(s), pork supply chains may choose the alternative(s) that best fit(s) their situation. In the event that surgical castration would someday be prohibited, derogations should be granted to production systems which have high constraints pertaining to a high slaughter weight, a high fat content of the products, or outdoor rearing. Tools for the objective examination of requests for such derogations are provided by the results of a study conducted within the EU-funded CASTRUM project [31].

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
