# Peer review of "Pros and Cons of Alternatives to Piglet Castration: Welfare, Boar Taint, and Other Meat Quality Traits"

_animals, 2019, doi:10.3390/ani9110884_

Round 1
Reviewer 1 Report
This very well written review focused on the alternative to surgical castration of male pigs. However, having read very carefully the manuscript, in my opinion, the specified objectives have not been met. This is a descriptive review where a critical discussion is missing. The data included in this manuscript are already known and published in other reviews as indicated by the authors. The novelty of this work should also be highlighted somehow. I suggest that the authors focus on recently published articles, ex from last 5-7 years. Right now the authors refer often to old publications.
Abstract
Line 21. Why increased unsaturation of the fat counts as a low quality? Nowadays consumers ask for low fat content and “healthier” fat.
Introduction.
Line 33. Currently many countries in EU and outside of EU practice castration with pain relief. Please comment on it, otherwise the reader can be misled.
Line 52. Change “Clasu” to “Claus”
Line 56. It should be mentioned already here that there are large variations in the sensitivity to andostenone.
Lines 59-61. Please rephrase this sentence so the reader can understand that androstenone can be reduced to alpha-androstenol and beta-androstenol in both liver and salivary glands.
Line 74. Delete “testosterone”
Line 75. Change “Wierrcinska” to “Wiercinska”
Line 76. [11]? Please check citation.
Line 97. [24]? Please check citation.
Section “8.3. Continuation of surgical castration”. This section can be expanded with references and more detailed information on reasons to continue surgical castration in different countries. It is a bit surprising to read that Italian stakeholders are very reluctant to abandon surgical castration because they need high quality dry cured products, and Eastern Europe just do not care about castration. I suggest to include reference here (especially for Eastern Europe). In Eastern European countries they have a traditional local product made from back fat of pigs. As in case with Italian products, the fat for such product should also be saturated.
Author Response
Comments and Suggestions for Authors
This very well written review focused on the alternative to surgical castration of male pigs. However, having read very carefully the manuscript, in my opinion, the specified objectives have not been met. This is a descriptive review where a critical discussion is missing. The data included in this manuscript are already known and published in other reviews as indicated by the authors. The novelty of this work should also be highlighted somehow. I suggest that the authors focus on recently published articles, ex from last 5-7 years. Right now the authors refer often to old publications.
Thank you for the kind and encouraging remarks. The castration issue has indeed a considerable history in European meat science, this is reflected by the in part old publications. As both authors are involved in this researchs since more than 30 years, it was important to us to show, that most of the basic knowledge in this field is available since decades, but the problems still are not solved. However more than 50% of the references are less than 5 years old.
Abstract
Line 21. Why increased unsaturation of the fat counts as a low quality? Nowadays consumers ask for low fat content and “healthier” fat.
Reference to more unsaturated fat being healthier has been added in line ???. For meat products, especially dry cured traditional products the sensoric quality is reduced by unsaturated fatty acidy. Thus meat industry in several countries prefers fat with a high amount of saturated fatty acids. These aspects were discussed in detail in the Castrum report, to which we refer several times.
Introduction.
Line 33. Currently many countries in EU and outside of EU practice castration with pain relief. Please comment on it, otherwise the reader can be misled.
A statement has been added to account for that.
Line 52. Change “Clasu” to “Claus”. Changed
Line 56. It should be mentioned already here that there are large variations in the sensitivity to andostenone. ”Sensitive” has been added before “consumers” in line ???
Lines 59-61. Please rephrase this sentence so the reader can understand that androstenone can be reduced to alpha-androstenol and beta-androstenol in both liver and salivary glands.
Line 74. Delete “testosterone”
Line 75. Change “Wierrcinska” to “Wiercinska” Changed
Line 76. [11]? Please check citation. Relevant citation added
Line 97. [24]? Please check citation. Relevant citation added
Section “8.3. Continuation of surgical castration”. This section can be expanded with references and more detailed information on reasons to continue surgical castration in different countries. It is a bit surprising to read that Italian stakeholders are very reluctant to abandon surgical castration because they need high quality dry cured products, and Eastern Europe just do not care about castration. I suggest to include reference here (especially for Eastern Europe). In Eastern European countries they have a traditional local product made from back fat of pigs. As in case with Italian products, the fat for such product should also be saturated.
Reviewer 2 Report
The paper is well written. At the beginning I had doubts about this paper because there are a lot of reviews of this issue, most of them cited by the authors. Moreover, some of the sections are quite superficially considered (i.e. 5.2.3), but I suppose it is not possible to talk about all the points deeply, so I think it is ok at least to name them and give little information as it is. However, the point of view of this paper and the message that gives I think it is very interesting and very important at that moment, at least in Europe, where the castration is being considered in several countries.
I do not have a lot of comments on the paper, just some small typing mistakes and some suggestions:
Line 52: Claus
Line 67: Is really a good reference for this sentence. I think this work is with trained panel.
Line 80: Maybe “welfare aspects of surgical castration”?
Line 97: Reference 24?
Line 105: Can the authors add a reference for the DFD problem in entire males?
Line 109-114: This summary include information from 3.1. and 3.2. I wonder if it should be in a separate section (summary of pros and cons of surgical castration…) or at least, to be separated from the previous paragraph
Line 131: “: “ instead of “;” after 2015
Section 5.1. : Should it be interesting to have a section about “How to deal with management problems”?
Line 164: A bracket is missing before Wesoly
Line 209: … and fat deposition
Line 231: Do the authors have a reference for this part?
Line 254 and following: References are missing
Line 273: Reference 59?
Line 279: Castrum
Lines 293 and 295: plus instead of +
Line 300: Commission
Line 308: Not only in high quality products, also in some bulk production.
Line 318: is instead of if.
Line 329: documented
Line 362: or economical benefits
Author Response
Comments and Suggestions for Authors
The paper is well written. At the beginning I had doubts about this paper because there are a lot of reviews of this issue, most of them cited by the authors. Moreover, some of the sections are quite superficially considered (i.e. 5.2.3), but I suppose it is not possible to talk about all the points deeply, so I think it is ok at least to name them and give little information as it is. However, the point of view of this paper and the message that gives I think it is very interesting and very important at that moment, at least in Europe, where the castration is being considered in several countries.
I do not have a lot of comments on the paper, just some small typing mistakes and some suggestions: Thank you for the kind remarks and your positive contributions.
Line 52: Claus Changed
Line 67: Is really a good reference for this sentence. I think this work is with trained panel.
Line 80: Maybe “welfare aspects of surgical castration”? Title was changed
Line 97: Reference 24? Changed to the relevant citation
Line 105: Can the authors add a reference for the DFD problem in entire males?
Line 109-114: This summary include information from 3.1. and 3.2. I wonder if it should be in a separate section (summary of pros and cons of surgical castration…) or at least, to be separated from the previous paragraph
A line was added to separate this section from 3.2
Line 131: “: “ instead of “;” after 2015 Changed
Section 5.1. : Should it be interesting to have a section about “How to deal with management problems”?
Information on how to deal with management problems is indeed valuable. A statement has been added at the end of section 5.1 to refer to European Commission 2019.
Line 164: A bracket is missing before Wesoly Bracket added
Line 209: … and fat deposition The increase in fat deposition is documented a few line below: “After that, fat development is greatly stimulated”
Line 231: Do the authors have a reference for this part?
Line 254 and following: References are missing
Line 273: Reference 59? Changed to the relevant reference
Line 279: Castrum Changed
Lines 293 and 295: plus instead of + Changed
Line 300: Commission Changed
Line 308: Not only in high quality products, also in some bulk production.
The sentence has been changed
Line 318: is instead of if. Changed
Line 329: documented Changed
Line 362: or economical benefits Economical benefits derive from the dimensions included in our statement
Round 2
Reviewer 1 Report
The authors have satisfied my concerns; the manuscript is now acceptable for publication.